# What is best practice in sex and relationship education? A synthesis of evidence, including stakeholders' views

Pandora Pound,[1] Sarah Denford,[2] Janet Shucksmith,[3] Clare Tanton,[4] Anne M Johnson,[5] Jenny Owen,[6] Rebecca Hutten,[7] Leanne Mohan,[3] Chris Bonell,[8] Charles Abraham,[9] Rona Campbell[1]

► Prepublication history and additional material are available. To view please visit the journal online ().

## ABSTRACT

**Objectives** Sex and relationship education (SRE) is regarded as vital to improving young people's sexual health, but a third of schools in England lacks good SRE and government guidance is outdated. We aimed to identify what makes SRE programmes effective, acceptable, sustainable and capable of faithful implementation.

**Design** This is a synthesis of findings from five research packages that we conducted (practitioner interviews, case study investigation, National Survey of Sexual Attitudes and Lifestyles, review of reviews and qualitative synthesis). We also gained feedback on our research from stakeholder consultations.

**Settings** Primary research and stakeholder consultations were conducted in the UK. Secondary research draws on studies worldwide.

**Results** Our findings indicate that school-based SRE and school-linked sexual health services can be effective at improving sexual health. We found professional consensus that good programmes start in primary school. Professionals and young people agreed that good programmes are age-appropriate, interactive and take place in a safe environment. Some young women reported preferring single-sex classes, but young men appeared to want mixed classes. Young people and professionals agreed that SRE should take a 'life skills' approach and not focus on abstinence. Young people advocated a 'sex-positive' approach but reported this was lacking. Young people and professionals agreed that SRE should discuss risks, but young people indicated that approaches to risk need revising. Professionals felt teachers should be involved in SRE delivery, but many young people reported disliking having their teachers deliver SRE and we found that key messages could become lost when interpreted by teachers. The divergence between young people and professionals was echoed by stakeholders. We developed criteria for best practice based on the evidence.

**Conclusions** We identified key features of effective and acceptable SRE. Our best practice criteria can be used to evaluate existing programmes, contribute to the development of new programmes and inform consultations around statutory SRE.

For numbered affiliations see end of article.

**Correspondence to**
Dr Pandora Pound;
pandora.pound@bristol.ac.uk

## Strengths and limitations of this study

► Our study involves both qualitative and quantitative research conducted in the UK, evidence syntheses that draw on data from all over the world and public involvement activities.
► The research data include the views of young people who have had sex and relationship education (SRE), as well as professionals involved in commissioning and delivering SRE.
► Triangulation of methods strengthens the validity of the criteria for best practice that we distilled from our synthesis of the individual research packages.
► A limitation of our study is that we did not investigate in depth the views and experiences of teachers who deliver SRE, nor did we manage to explore the views of parents.
► Although the international data support our UK-based findings, our criteria for best practice probably have greatest relevance in UK settings.

towards sexuality, greater variation in sexual behaviour and increased gender equality.[1] Meanwhile, new digital technologies and widespread internet access have affected the ways in which young people learn about sex and conduct their sexual lives, bringing new opportunities but also new risks for young people.[2 3] School-based sex and relationship education (SRE) is seen as vital for navigating these changes, safeguarding young people[4] and helping to combat child sexual abuse[5] and exploitation.[6] Despite this, more than a third of schools in England lacks good-quality SRE,[7] and there are concerns about disparities in the content and quality of provision,[7] disparities that might partly explain the social inequalities observed in sexual health.[8] Government guidance on SRE is now 17 years old,[9] and to date the absence of statutory SRE (apart from the element in the Science National Curriculum) has meant that each school develops its own approach. However, the government has recently announced its intention to make SRE statutory in all

## BACKGROUND

Young people find themselves in a shifting sexual landscape due to changing attitudes

secondary schools (maintained, independents and academies) and to make relationships education statutory in all primary schools.[10]

SRE also represents a key strand in policies to improve sexual health outcomes among young people.[11 12] Although the under 18 conception rate in England and Wales is currently at its lowest since 1969,[13] pregnancies in women aged 16–19 are commonly unplanned[14] and the rate remains high compared with the rest of Western Europe.[15 16] UK rates of sexually transmitted infections (STIs) also remain relatively high,[17] with 16-year-olds to 24-year-olds accounting for most new UK diagnoses despite comprising only 12% of the population.[18] There is also evidence that young people are particularly vulnerable when it comes to unwanted sexual experiences; the median age for non-volitional sex is 18 (women) and 16 (men),[19] and the reporting of sexual offences against children and young people is increasing in the UK.[20]

Our long-term goal is to either develop an evidence-based universal SRE intervention for use in English secondary schools, or to adapt and evaluate a suitable existing intervention. The study we report here is preliminary to this long-term goal; it aimed to gather evidence about best practice in SRE and to identify characteristics that make SRE programmes effective, acceptable, sustainable and capable of implementation with fidelity.

## METHODS

We conducted five pieces of research, several of which have already been published separately. We synthesised the findings from each of these studies to distil the evidence relating to best practice. A summary of the methods and findings for each of the individual pieces of research is shown in online supplementary appendix 1, and more details are available in the published papers. The aspects addressed in each of the studies (acceptability, sustainability, etc) are shown in table 1. In addition to the five studies, we consulted stakeholders for their views on our research.

## Work packages

### Telephone interview study with practitioners in local authorities across all English regions to investigate best practice and obstacles to this

Purposive and snowball sampling techniques were used to identify key contacts in local authorities. Of 61 local authorities contacted, 36 agreed to take part, resulting in 39 interviews conducted with 40 individuals (one interview was paired). Additionally seven interviews were conducted with individuals from six purposively sampled relevant national organisations. In total 47 individuals were interviewed, with representation from every English region, although we were only able to recruit one participant from the West Midlands. Most local authority interviewees were service commissioners, 'Healthy Schools' coordinators, education advisers supporting Personal, Social and Health Education (PSHE), or child and young people's health specialists. Most interviewees were based in metropolitan areas and unitary authorities, although some county councils with large rural populations were also represented. Interviews were digitally recorded, transcribed and analysed using the 'framework' method.[21]

### Synthesis of qualitative studies of young people's views of their SRE

Studies were located using comprehensive electronic and hand-searching methods. References identified through electronic databases were double-screened, and eligible papers were appraised for quality by two independent reviewers. Studies were included if they involved young people aged 4–19 in full-time education, young adults ≤19 or adults ≤25 if recalling their SRE, if they employed qualitative methods of both data collection and analysis, and explored school-based education delivered by teachers, peer or external educators. Studies were excluded if they focused solely on alcohol or HIV/AIDS, were conducted before 1990 or involved special schools or students with special needs. Sixty-nine publications were identified, with 55 remaining after quality appraisal, representing 48 studies spanning 25 years. Only one study involved

| Work package | Study methodology | Aspects addressed |
|---|---|---|
| | **Table 1** Aspects of SRE programmes addressed by the five studies | |
| 1 | Telephone interview study with practitioners in local authorities across all English regions to investigate best practice and obstacles to this | Effectiveness (practitioners' views of), sustainability |
| 2 | Synthesis of qualitative studies of young people's views of their SRE | Acceptability |
| 3 | Case study investigation of factors that make interventions acceptable to young people, parents and those delivering them | Acceptability, fidelity, sustainability |
| 4 | Exploration of data from the third National Survey of Sexual Attitudes and Lifestyles | Effectiveness, acceptability |
| 5 | Review of systematic reviews of school-based sexual health and alcohol interventions | Effectiveness |

SRE, sex and relationship education.

primary school children. Most were of standard SRE delivered to secondary-school pupils by teachers, using a focus group methodology. Studies came mainly from the UK, followed by the USA, New Zealand, Canada, Ireland, Australia, Japan, Iran, Brazil and Sweden. A meta-ethnographic approach was taken to data analysis. Full details of this study have been reported elsewhere.[22]

### Case study investigation of factors that make interventions acceptable to young people, parents and those delivering them

The aim was to examine distinct models of intervention informed by different theories of behaviour change. A scoping exercise identified the range of interventions currently embedded in English schools and three diverse interventions were purposively sampled: a social norms approach, a curriculum-based risk and resilience model, and a comprehensive, school-based peer education programme provided by a third-sector organisation. Relevant documents were examined for each of the programmes. The number and range of participants varied for each of the three interventions — *social norms intervention*: interviews with four members of the local authority team involved in intervention design and delivery, interviews with three school teaching staff, and observation of intervention delivery and feedback sessions; *risk-taking behaviour toolkit*: four single-sex focus groups with five young people each, interview with PSHE lead, and informal discussions with SRE coordinator involved in intervention development, with Education Improvement Service team and with tutors in the school; and p*eer education programme*: interviews with the project team and two volunteers. Parents were invited to participate in focus groups but none accepted. Data were analysed qualitatively and a cross-case analysis was conducted.

### Exploration of data from the third National Survey of Sexual Attitudes and Lifestyles

Data from National Survey of Sexual Attitudes and Lifestyles (Natsal-3) were analysed to investigate (1) associations between sources of information about sex and sexual health outcomes, and (2) trends in sources of information about sex and information needs among young people. Full details of these analyses have been reported elsewhere.[23 24] The Natsal probability sample surveys to date have been carried out approximately decennially in 1990–1991 (Natsal-1), 1999–2001 (Natsal-2) and 2010–2012 (Natsal-3). In all three surveys, households were selected using stratified probability sampling, from which one eligible individual, resident in Britain, was selected at random and invited to participate. Natsal-3 interviewed 15 162 men and women aged 16–74 years (1729 men and 2140 women aged 16–24 years). The overall response rate was 57.7%. Participants were interviewed using computer-assisted personal interviewing, with computer-assisted, self-interview for the more sensitive questions. Analyses for questions (1) and (2) above were restricted to those aged 17–24 years and 16–24 years at interview, respectively.

As part of question (2), participants were given a list of topics and were asked 'Looking back to the time when you first felt ready to have some sexual experience yourself, is there anything on this list that you now feel you ought to have known more about'.

### Review of systematic reviews of school-based sexual health and alcohol interventions

The Cochrane Database of Systematic Reviews, Embase, MEDLINE, MEDLINE In-Process, Web of Science and PsycINFO were searched. Titles and abstracts were screened and potentially relevant articles retrieved in full. An independent reviewer checked a random 10% sample of titles and abstracts as well as full papers. Reviews were eligible if they were systematic reviews or meta-analyses of randomised controlled trials, cluster randomised trials or studies using a quasi-experimental design, and of interventions targeting those aged 4–19 years in full-time education, school-based sexual-health interventions, school-linked sexual health services, interventions combining alcohol use and sexual health education, or interventions to combat multiple risk behaviours. Reviews were excluded if their primary focus was sexual health screening, sexual assault or abuse, rape prevention or children with developmental disorders. Review quality was assessed using assessing the methodological quality of systematic reviews (AMSTAR),[25] after which 37 systematic reviews of 224 studies were included. Interventions evaluated in these studies included abstinence-only programmes, comprehensive programmes, pregnancy prevention programmes, HIV prevention programmes, and school-based or school-linked clinics. Studies came from the USA, Canada, Australia, New Zealand, Europe, Africa, Asia and South America. A narrative synthesis was conducted. Full details have been reported elsewhere.[26]

### Stakeholder consultations

We consulted stakeholders to gain their feedback on our research, specifically in areas where the research had indicated uncertainty, lack of consensus or lack of evidence. Consultations were held with three young people's groups, in Cardiff, Bristol and Newcastle, totalling 55 young people (of whom 35 were girls) between the ages of 11 and 18. On each occasion we presented our findings and then held discussions, which were either transcribed (one consultation) or recorded using notes (two consultations). In one consultation we also collected young peoples' feedback using a short questionnaire. We were particularly interested in young people's views on who should deliver SRE, whether lessons should be mixed-sex or single-sex and how to improve SRE. Young people's feedback was organised thematically and summarised in a report.[27]

Our consultation with professionals consisted of a 1-day workshop in London with 19 experts and practitioners working in the field of SRE, where we discussed the evidence following a presentation of our research

and worked on selected issues in small groups. We asked professionals for their views on the sort of approach SRE programmes should take, when SRE should start, whether lessons should be mixed-sex or single-sex, who should deliver SRE and what they felt about engaging with parents over SRE. Because participants in public involvement activities do not consent to have their contributions treated as research data, we did not collect detailed information on our stakeholders, nor did we analyse their feedback as research data; we were simply interested in their views on our research and on our evidence gaps. Professionals' feedback was organised thematically and summarised in a report.[28]

## Synthesis

The evidence generated from the five work packages was synthesised and is presented below. Data from the stakeholder consultations are presented separately, after the research evidence. (Online supplementary appendix 1 provides a summary of the methods and findings for each of the individual research packages.)

## RESULTS
### Findings from work packages
#### SRE in schools

Data from Natsal-3 suggest that schools are important sources of information about sex for young people. During the period 1990–2012 the proportion of 16-year-olds to 24-year-olds who cited school lessons as their main source of information about sex increased from 28% to 40%, while the proportion of those who felt they ought to have known more (when they first felt ready for some sexual experience) was lowest among those reporting school as their main source of information. Among those who wanted to know more, school was the most commonly reported preferred source for both males and females.

#### School-based SRE is associated with positive reported outcomes

Data from Natsal-3 indicate that young people who report lessons at school as their main source of information about sex are less likely to have had unsafe sex in the past year than young people who report receiving most of their information about sex from other (non-parental) sources. Those who learn about sex mainly through school lessons also tend to report being older the first time they have sex and are less likely to report having had an STI diagnosis. Women who report receiving most of their information about sex from school-based sex education are more likely to report being 'sexually competent' the first time they have sex (ie, both partners are 'equally willing', reliable contraception is used, the decision to have sex is not due to peer pressure, drunkenness or drugs, and sex occurs at the perceived 'right time') and less likely to report having experienced non-volitional sex, abortion or distress about sex.

#### Effectiveness of different SRE programmes

The review of reviews suggests that comprehensive programmes (which aim to prevent, stop or decrease sexual activity, but also promote condom use and other safer sex strategies) can be effective at improving knowledge, skills and attitudes. HIV prevention programmes appear to be effective at increasing knowledge. Pregnancy prevention programmes appear effective at improving knowledge and those targeting social disadvantage may reduce teenage pregnancies. Abstinence-only programmes are not effective at promoting positive changes in sexual behaviour. There was limited evidence on programmes that target both alcohol use and sexual behaviour simultaneously.

#### Effectiveness of school-based or school-linked sexual health services

School-based or school-linked sexual health services appear to be effective at reducing sexual activity, numbers of sexual partners and teenage pregnancies, according to the review of reviews. Professionals participating in the interview study suggest that best practice in SRE involves close liaison with relevant sexual health and advice services, either through offering school-based services or through links with local sexual health services. The qualitative synthesis found that the few young people who attended a school-linked clinic for SRE were enthusiastic about this and appreciated the educator's expertise and enthusiasm.

#### Professionals' and young people's views on different approaches to SRE
##### 'Whole school' ethos

There was consensus among professionals involved in both the interview study and the case study investigation that SRE should be integrated into a 'whole school' ethos; that is, SRE should take place within a school context that promotes and embodies a consistent set of principles and values (eg, the promotion of respectful interactions) within both formal and informal practices.

##### Life skills approach

There was consensus among professionals participating in the interview study that good SRE should promote resilience and teach life skills, for example planning and communication skills, decision-making skills and how to assess risks and resist 'peer pressure'. The qualitative synthesis found that young people wanted to learn refusal skills and become more confident in sexual negotiations. In Natsal-3 11% of young men and 17% of young women stated that they wanted more information on refusal skills.

##### Risky behaviour approach

There was strong consensus among professionals involved in both the interview study and the case study investigation that topics such as sexual health and alcohol should be integrated into a 'risky behaviour approach', whereby different behaviours are discussed both as distinct domains and in terms of how they relate to each other. Just under half of young people in Natsal-3 reported

wanting more information on at least one 'risk-reduction topic', for example STIs and contraception. However, the qualitative synthesis found that young people dislike SRE that emphasises the risks and negative aspects of sexual activity without acknowledging the positive and pleasurable aspects. As a boy reported in a study[29] included in the qualitative synthesis, "All they ever do is talk about the dangers of sex and that, and nothing about the pleasure." Young people's accounts also suggest that common approaches to sexual risk-taking within SRE do not resonate with their experiences; some young people perceive the risks of unsafe sex to be less than those potentially entailed by safer sex (eg, loss of erection due to discussing condoms during sex). The qualitative synthesis suggests that approaches focusing on risk-taking need to be developed in careful consultation with young people.

### Abstinence approach

There was consensus among professionals in the case study investigation that SRE messages need to be more positive about sex than the traditional abstinence approaches. In the qualitative synthesis many young people reported disliking the emphasis on abstinence within SRE, finding it unrealistic.

### 'Sex-positive' approach

Two of four programmes considered by professionals (in the interview study) to represent good practice employed a 'sex-positive' approach, and the qualitative synthesis suggests that young people's aspirations for SRE also accord closely with such an approach. (There is no single definition of 'sex-positive', but broadly it is an approach that is open, frank and positive about sex, that challenges negative societal attitudes to sex and that embraces sexual diversity at the same time as emphasising the importance of consent and comprehensive SRE.) Twenty per cent of young men and 17% of young women in Natsal-3 reported wanting to know more about how to make sex more satisfying, while some young people in the qualitative synthesis also reported wanting to learn about the pleasurable aspects of sex. In the qualitative synthesis young people expressed a desire for more openness in SRE and more discussion about what sex involves and how to have sex. They wanted SRE to be more relevant, to reflect that some young people are sexually active and to acknowledge their autonomy and maturity. Additionally they expressed a need for accurate, relevant and unbiased information, for example about abortion, different forms of contraception or the options available in the event of pregnancy. Young people in both the qualitative synthesis and Natsal-3 (23% of young men and 29% of young women) wanted SRE to include discussions of relationships and of the emotions and feelings that can accompany sexual activity. As one young person reported in one of the qualitative synthesis studies,[30] "They don't really go into the whole relationships thing partly because I don't think - they don't want us to have relationships."

Young people also commonly observed that SRE was gendered and heterosexist. Some reported (in the qualitative synthesis) that lesbian, gay, bisexual and transgender students were invisible within SRE. Young people in both Natsal-3 (8.5% of both young men and women) and the qualitative synthesis expressed a wish for information on, and discussion of, same-sex relationships. In Natsal-3 around 9% reported wanting more information on a range of sexual practices, including masturbation (7%–8%), while those in the qualitative synthesis reported that SRE defined sex narrowly as heterosexual intercourse and ignored the range and diversity of the sexual activities they engaged in. A young person reported in a study[31] included in the qualitative synthesis put it like this: "So you just were taught about sexual intercourse causing pregnancy, but you were never taught about masturbation; you were never taught about oral sex all the different, other types of sexual practices." Young people reported that SRE also failed to discuss female sexual pleasure, reproduced stereotypes of women as passive and lacking in desire, placed responsibility for the work of sexual relationships onto women and cast women in the role of sexual gatekeepers.

### Culturally sensitive approach

Young people in the qualitative synthesis reported that SRE could occasionally be culturally insensitive. Nevertheless some pupils from ethnic or religious minorities valued SRE because sex was not discussed within their families and/or because it challenged the information they received at home.

### Characteristics of good SRE provision
### Adaptable

Professionals involved in both the interview study and the case study investigation expressed their view that — within the context of comprehensive SRE — programmes need to be adaptable to different school environments, structures, timetables and class groupings, with the content sufficiently flexible for it to be easily and immediately adapted according to local need. In the case study investigation they suggested that the core and peripheral features of the curriculum be clearly identified so that the main elements are always covered and a 'minimum dose' ensured.

### Age-appropriate

The review of reviews concluded that programmes should be appropriate for participants' culture, age and sexual experience. In the qualitative synthesis young people reported that SRE is delivered too late. Programmes identified as good by professionals in the interview study were those that delivered SRE from primary school onwards.

### Spiral

Professionals in the interview study believed that best practice should involve a 'spiral' curriculum with age-appropriate stages delivered via regular lessons, as well as special projects and events. As a spiral curriculum involves returning to

the same topics to reinforce learning, professionals in the case study investigation noted the importance of ensuring progression and avoiding inappropriate repetition if young people are to feel they are progressing.

### Of sufficient duration and intensity

The review of reviews concluded that programmes should be of sufficient duration and intensity. Professionals involved in both the interview study and the case study investigation felt that the use of single 'drop down days' (where a whole SRE programme is delivered in 1 day) was poor practice if they constituted the only SRE provision within a school. As a respondent in the case study investigation put it: "Research has shown that timetabled regular PSHE is a lot better than not doing anything and then once every term or every half term having a one day. Because what if a child's absent that day? Then they're not getting anything, and actually, you know, by the very nature [of it] PSHE is something that you need to be practising on, building on those skills that you're talking about, having those discussions around values and around attitudes and the knowledge, you know, and having those scenarios to be able to practice those skills with." However, if drop-down days supplemented an ongoing programme, they were felt by professionals in the case study investigation to potentially bring young people into valuable contact with external educators.

### Interactive and engaging

The review of reviews concluded that programmes should employ interactive and participatory educational strategies that actively engage recipients. The qualitative synthesis indicated that young people appreciate interactive, dynamic teaching techniques and want SRE to include group discussions, skills-based lessons, demonstrations and diverse activities.

### Safe

The review of reviews concluded that SRE programmes should create a safe environment for young people. Young people in the qualitative synthesis agreed; they wanted SRE to take place in an environment where they could participate uninhibitedly without concerns about being singled out or ridiculed. They commonly reported high levels of discomfort, particularly in mixed-sex classes, with young men and women both feeling vulnerable for different reasons. Some young women reported being verbally harassed by young men if they engaged in the class, while young men's frequently reported disruptive behaviour was interpreted as an attempt to prevent exposure of any sexual ignorance. Some young women and girls expressed a preference for single-sex classes all or some of the time, but young men appeared to want mixed-sex classes. Young people advocated small group teaching or smaller classes that were deemed easier to control. They considered good class control to be essential for creating safety in SRE.

### Confidential

The qualitative synthesis found that building trust between classmates could increase engagement in SRE, while ground rules (for discussion, behaviour and confidentiality) could reduce discomfort. It also indicated that educators who were separate from the school might enhance young people's trust that confidentiality would be maintained. The case study investigation suggested that distancing techniques (ie, discouraging young people from discussing personal issues for the purposes of maintaining confidentiality) could lead to some young people disengaging as they did not find it meaningful to discuss fictional accounts. As one young person reported in the case study investigation, "Instead of it just being like when they tell you stories that are probably made up, like off the internet or something and they tell you all these stories and you're like, but I don't care, I don't know who the person is."

### Who should deliver SRE?

Young people report, in the qualitative synthesis, that good sex educators enjoy teaching SRE, have experiential knowledge and are comfortable with their own sexuality. They are professional, confident, unembarrassed, straightforward, experienced at talking about sex and use everyday language. They have expertise in sexual health, are specifically trained in SRE, are trustworthy, approachable, non-judgemental and able to maintain confidentiality. They respect young people and their autonomy, treat young people as equals and accept that they may be sexually active.

### School teachers

Most professionals in the interview study acknowledged that current teacher training did not prepare teachers to deliver SRE, but they nevertheless emphasised a need for teachers to be involved in its delivery. They suggested that good practice involved a partnership between teachers and others, such as school nurses or experts, from specialist third-sector organisations. The qualitative synthesis found, however, that young people generally regard school teachers as unsuitable for delivering SRE, perceiving them to be inadequately trained, embarrassed and unable to discuss sex frankly, and judgemental and unaccepting of young people's sexual activity.

Additionally young people in the qualitative synthesis frequently commented that having SRE delivered by familiar teachers with whom they had an ongoing relationship was 'awkward' and could compromise confidentiality. As a young person reported in one of the qualitative synthesis studies,[32] "not teachers because [teachers] know you, judge you, and they like to talk about you." Some pupils in the case study investigation were taught SRE by their form tutors because the school felt they had developed strong bonds with these tutors. However, young people reported being particularly uncomfortable in these classes, suggesting that the discussion of sensitive topics with their form tutors involved a blurring of

boundaries. Young people in the qualitative synthesis also reported that delivery by teachers could blur boundaries and disrupt established relationships. Additionally their accounts suggested that the power imbalance inherent in the student–teacher relationship could be problematic within the context of SRE.

### Sexual health professionals

The qualitative synthesis indicates that many young people appreciate sexual health professionals visiting schools to teach SRE. Such professionals were perceived to be less judgemental, to 'know what they're talking about' and to be better at delivering SRE than teachers, although maintaining discipline was occasionally reported to be problematic. In both the qualitative synthesis and the case studies, young people reported that outside experts provide greater confidentiality and reduce discomfort and embarrassment due to their separateness from the school. The case study investigation found that sexual health professionals, local authority staff and young people all felt that external experts provided a higher quality of delivery because they were trained, their delivery involved no loss of programme fidelity, and because they were able to provide clear boundaries and afford young people a higher level of confidentiality.

### Peer educators

In the qualitative synthesis young people reported mutually respectful relationships and a sense of affinity with peer educators, claiming that peer-led SRE had an impact on them. They reported that their lack of prior relationship with peer educators made them easier to trust than teachers, although some feared that they might not take confidentiality seriously enough. Peer educators were considered by most young people (in both the qualitative synthesis and the case study investigation) to be highly credible, although some in the qualitative synthesis felt credibility could be undermined by youth or lack of knowledge.

### Factors to consider when developing new SRE programmes: preparation, fidelity and evaluation

Both the review of reviews and the interview study concluded that SRE programmes need to be carefully planned using logic models and developed with input from stakeholders, including young people. Professionals in both the interview study and the case study investigation recommended that SRE programmes are adequately resourced, have the support of the head teacher, are tailored to local needs and congruent with the values of the school. While it was felt that the whole school community should be prepared for new programmes, there was some indication that schools were cautious about engaging with parents over SRE.

The case study investigation found that key messages intended by SRE programmes could get lost when programmes were interpreted by teachers, for example messages might lack consistency, or more negative messages might be delivered than the approach intended. In addition

to lack of fidelity, both the case studies and the interview study concluded that a variety of other factors could affect programme implementation, including pressures on the curriculum, funding pressures, staff capacity, academic pressures, government policies (eg, the shift to academies) and the present lack of statutory status for SRE.

The review of reviews concluded that interventions should be evaluated according to both short-term and long-term outcomes, while professionals in the interview study recommended that health outcomes should be used, as well as school-relevant outcomes such as attainment and attendance. The qualitative synthesis indicated that young people should be involved in evaluations.

### Findings from stakeholder consultations
#### Consultation with young people's groups

Members of the young people's consultation groups agreed with the research evidence on most issues. With respect to the 'awkwardness' and blurring of boundaries involved in having SRE delivered by familiar school teachers, their views echoed the research evidence, with the important caveat that this issue might be specific to older pupils and not relevant to primary school children who were felt to possibly prefer familiar teachers. Similarly group members proposed that while mixed-sex classes might be preferable for secondary school pupils, primary school children might feel more comfortable in single-sex classes. Female group members appeared to be more interested than male group members in having a *combination* of single-sex and mixed-sex classes. The young people's groups also highlighted the need for SRE content to be updated to include teaching on consent, sexting, cyberbullying, online safety, sexual exploitation and sexual coercion.

#### Consultation with experts and practitioners working in the field of SRE

Professional stakeholders agreed with the research evidence in terms of the sort of approach SRE should take, but added that SRE should be integrated into the school ethos from nursery or reception classes onwards. They proposed that SRE should build up incrementally with age-appropriate topics, language and activities, and continue throughout the period of compulsory schooling, ideally up to age 18. Professionals also recommended a proactive approach to engaging with parents about SRE and suggested that school staff need to develop a clear position and language for talking to parents. However, while acknowledging the research evidence that young people could be vulnerable in mixed-sex classes, most professionals nonetheless felt it important that young men and women should learn together. Similarly they strongly disputed research findings indicating that young people dislike being taught by familiar teachers and were strongly of the opinion that the only long-term, sustainable option was for teachers to be involved in SRE delivery. Most felt that young people's concerns could be easily resolved by training teachers, adequately resourcing SRE, achieving statutory status and establishing boundaries for pupils before lessons.

## Box 1  Criteria for best practice in SRE

**Sexual health and advice services**
► SRE programmes should involve close liaison with relevant sexual health and advice services, either through school-based services or through links with local sexual health services.

**SRE curriculum model**
► SRE should be appropriate for pupils' culture, age and sexual experience. It should start in primary school and use age-appropriate language, topics and activities.
► SRE should continue throughout the period of compulsory schooling, ideally up to age 18.[*]
► SRE programmes should be of sufficient duration and intensity; that is, teaching should be delivered via regular lessons, as well as special projects and events. 'Drop down days' are only acceptable if they supplement an ongoing programme, not if they constitute the only SRE provision within a school.
► SRE curricula should be adaptable and flexible, and identify core and peripheral features.
► SRE programmes should use a spiral curriculum model, exploring topics in logical sequence and avoiding inappropriate repetition.
► Educators should employ a diverse range of interactive and participatory educational strategies and activities that actively engage recipients.
► Schools should take a proactive approach to engaging with parents about SRE.[*]

**SRE content**
► Bearing in mind age appropriateness, SRE should be 'sex-positive'; that is, it should be open, frank and informative, and should acknowledge the pleasures of sex. It should reflect that some young people are sexually active and acknowledge young people's autonomy and level of maturity. It should not focus on abstinence.
► SRE should reflect sexual diversity. It should discuss a range of sexual activity (not just heterosexual intercourse), as well as lesbian, gay, bisexual and transgender issues and relationships.
► SRE should include teaching on consent, sexting, cyberbullying, online safety, sexual exploitation and sexual coercion.[*]
► SRE should challenge, rather than reinforce, gender stereotypes and inequalities.
► SRE should be culturally sensitive.
► SRE should be integrated into a 'whole school' ethos and should teach life skills (eg, planning, decision-making skills), specific skills (eg, communication, sexual negotiation skills) and promote resilience.
► SRE should provide impartial information on contraception, safer sex, pregnancy and abortion.
► SRE should discuss relationships and emotions.
► Where appropriate, potentially risky practices should be considered in combination, for example considering the risks of sexual activity alongside substance use.
► Lessons on the risks of sexual activity need to be developed carefully; an overemphasis on risk can alienate some young people, particularly if the risks are emphasised at the expense of the positive and pleasurable aspects of sex.
► SRE programmes should be developed with input from young people.

**SRE delivery**
► SRE should take place in a safe environment for young people. This necessitates excellent class control and protection of students from harassment.
► Teaching should be delivered in small groups where appropriate and in single-sex groups at least some of the time. Primary school children may feel more comfortable in single-sex classes.[*]
► SRE should take place in a confidential environment. Distancing techniques should be used with caution to avoid student disengagement. Young peoples' trust in confidentiality is enhanced by the educator's separateness from the school.
► Staff delivering SRE should be trained educators, have expertise in sexual health, be sex-positive and enthusiastic about delivering SRE.
► External sexual health professionals should be involved in delivering SRE.
► School teachers delivering SRE should be willing to work in partnership with external sexual health professionals.
► Ideally staff delivering SRE to secondary school[†] pupils will not be in an ongoing relationship with students in another capacity (ie, will not be familiar to students as form or subject teachers). This is to protect student confidentiality, privacy and boundaries.
► External, trained peer educators have a role to play in delivering SRE, in partnership with expert educators.

*This criterion comes from stakeholder consultations; it does not constitute research evidence.
†Stakeholder consultations suggest that primary school-aged children might feel more comfortable with familiar teachers; however, this is only suggestive and does not constitute research evidence.
SRE, sex and relationship education.

## Criteria for best practice in SRE

On the basis of our research findings and consultations with stakeholders, we developed a list of criteria for best practice in SRE (box 1).

## DISCUSSION

We found clear evidence that school-based SRE and school-based or school-linked sexual health services can be effective in terms of improving both objective and reported sexual health outcomes. We found professional consensus that good SRE programmes start in primary school, are adaptable and employ a spiral curriculum model. Professionals and young people are in agreement

that good SRE programmes should be age-appropriate, interactive and engaging, and take place in a safe and confidential environment. Some young women and girls express a preference for single-sex classes all or some of the time, but young men appear to want mixed classes. Young people and professionals agree that SRE should take a 'life skills' approach and not focus on abstinence. Young people advocate a sex-positive approach but report that this is lacking at present. Young people and professionals agree that SRE needs to include discussion of risks, but young people's accounts indicate that current approaches to discussions of risk may need revising. Professionals believe that teachers have a key role to play in delivering SRE, but many young people dislike having their own teachers deliver SRE, and we found that key messages intended by SRE programmes can become lost or more negative when interpreted by teachers. Stakeholder consultations echoed the divergence between the views of young people and professionals.

This study's strength is that it synthesises findings from both qualitative and quantitative primary research and evidence syntheses, with young people's voices forming a strong part of the evidence. While each of the constituent research packages provides valuable findings in its own right, the synthesis delivers more than the sum of its parts, with the studies complementing and supporting each other's findings. The qualitative studies offer meaning and explanation for the quantitative findings, while the latter support and strengthen the qualitative findings. The criteria generated from this research are likely to be of use to practitioners, commissioners and researchers in the field of SRE. They are likely to be of particular value during the forthcoming consultations to draw up guidance for statutory SRE in all English schools.[10] The study also benefits from the discussions we held with stakeholders about our findings, which we found essential for keeping our recommendations grounded and realistic. However a limitation of the study is that, although a small number of teachers were interviewed as part of the case study investigation, we did not investigate teachers' views in greater depth. Additionally, the case study investigation was unsuccessful in its attempts to recruit parents so their views are not represented here. Finally, although the evidence syntheses drew on data from all over the world, our primary research data are confined to the UK. Although the international data support and do not contradict our UK-based findings, our criteria for best practice probably have greatest relevance in a UK setting.

Our study draws attention to the importance of focusing on SRE delivery as well as content. Some recommended aspects of delivery are relatively easy to put into practice, for example ensuring pupils feel comfortable and safe by offering a combination of single-sex and mixed-sex classes, or by maintaining excellent class control in SRE. Similarly, recommendations concerning the structure and content of the curriculum may be relatively easily implemented by schools; however, the real challenge lies in *delivering* the curriculum appropriately. For example,

excellent information might be relatively easily imparted in lessons but is much more challenging for schools to back this up by developing strong links with sexual health services, or by implementing school-based sexual health services. Similarly, an excellent sex-positive curriculum may be developed, but unless it is delivered by an enthusiastic, experienced and sex-positive educator its key intended messages run the risk of becoming lost.

In terms of who should deliver SRE, young people emphasised the need for acceptability while professionals highlighted the importance of sustainability. We were surprised at how robustly professionals in our stakeholder consultation challenged the evidence about young peoples' dislike of their own teachers delivering SRE. Of course research evidence is only one of many types of knowledge that can be applied in practice,[33–35] and some practitioners may place a higher value on experiential knowledge,[36] but there is a risk that young people will disengage from SRE if their concerns about educators are not adequately addressed. While their accounts may have been based on experiences of poor teaching, their views should not be dismissed on this basis; better training is not the only solution since many of young people's concerns relate to the student-teacher relationship in the context of SRE, not their teachers' pedagogic skills. On the other hand external sexual health professionals meet many of young people's criteria for acceptability but potentially pose a problem of sustainability, at least within the current context of economic austerity.

Future studies might explore the acceptability and sustainability of different models of delivery. The coteaching model, whereby sexual health professionals collaborate with teachers on an ongoing basis to deliver SRE, appears to be appreciated by both students and teachers where employed.[37] However if SRE is to continue being delivered predominantly by teachers, one possibility would be for secondary schools to have a dedicated SRE teacher who *only delivers SRE* (possibly also to neighbouring schools on a peripatetic basis). Such a person could potentially offer expertise, confidentiality and continuity, but also distinct boundaries since their only relationship with students would be as their SRE teacher. Further research could also investigate teachers' views and experiences of delivering SRE. The available evidence suggests that they are often uncomfortable delivering SRE,[38–42] that many lack confidence teaching the subject[43–45] and that only very few feel they should be the sole providers of SRE.[46] However they appear to hold mixed views about the impact of teacher-delivered SRE on the teacher–student relationship.[39 42] In general teachers' accounts point to the extraordinary challenges involved in discussing sex within an environment that strives to be desexualised.[47]

## CONCLUSIONS

We conducted and synthesised a wide range of research and stakeholder consultations to identify what makes SRE programmes effective, acceptable, sustainable and

capable of faithful implementation. Our findings highlight the importance of focusing on SRE delivery as well as content. We uncovered a divergence between the views of young people and professionals on how to deliver SRE, a divergence that reflects potential conflict between the principles of acceptability and sustainability. Nevertheless we generated criteria for best practice based on the evidence. These criteria will be of value to those interested in developing high-quality SRE programmes to help safeguard young people and improve their sexual health. We hope that they will also help to inform the forthcoming consultations around developing guidance for statutory SRE in English schools.

**Author affiliations**
[1]School of Social and Community Medicine, University of Bristol, Bristol, UK
[2]Children's Health and Exercise Research Centre, University of Exeter St Luke's Campus, Exeter, UK
[3]School of Health and Social Care, Health and Social Care Institute, Teesside University, Middlesbrough, UK
[4]Research Department of Infection and Population Health, Institute of Epidemiology and Health Care, University College London, London, UK
[5]Institute of Epidemiology and Health Care, University College London, Institute of Epidemiology and Health Care, London, UK
[6]Public Health Section, School of Health and Related Research (SHARR), University of Sheffield, Sheffield, UK
[7]School of Health and Related Research (SHARR), University of Sheffield, Sheffield, UK
[8]Department Social and Environmental Health Research, London School of Hygiene and Tropical Medicine, London, UK
[9]Psychology Applied to Health, University of Exeter, Exeter, UK

**Contributors** PP conducted the searches, quality appraisal, data extraction and qualitative synthesis for Work Package 2. She conducted stakeholder consultations with young people and experts and practitioners, synthesised the evidence from the five work packages and from stakeholders, and drafted this paper. RC conceived the idea and design for the overall project, as well as the qualitative synthesis (Work Package 2). She conducted the stakeholder consultation with experts and practitioners, contributed to the interpretation of the qualitative synthesis, the design and reporting of Work Package 5, and assisted in drafting and critically revising this paper before approving the final version. SD conducted the searches, quality appraisal, data extraction and narrative synthesis for Work Package 5, contributed to the stakeholder consultation with experts and practitioners, and assisted in critically revising this paper before approving the final version. JS contributed to the design for the overall project and designed and managed Work Package 3. She contributed to the stakeholder consultation with experts and practitioners, the synthesis of the data, and assisted in critically revising this paper before approving the final version. LM collected and analysed data for Work Package 3. JO and RH planned and carried out the interviews for Work Package 1. They both conducted preliminary analyses of the data, and JO compiled a report that informed this paper. JO contributed to the stakeholder consultation with experts and practitioners. AMJ contributed to the analysis of Natsal-3 (Work Package 4), critically revised this paper and approved the final version. She contributed to the stakeholder consultation with experts and practitioners. CT led one of the analyses of the Natsal-3 data for Work Package 4 and contributed to the other analysis in Work Package 4. She critically revised this paper and approved the final version. CB contributed to the interpretation of study findings, the stakeholder consultation with experts and practitioners, and the drafting of the paper. CA worked on the design, execution and reporting of Work Package 5, and commented on the draft paper. He contributed to the stakeholder consultation with experts and practitioners.

**Funding** This paper presents independent research funded by the National Institute for Health Research School for Public Health Research (NIHR SPHR). The views expressed are those of the authors and not necessarily those of the NHS, the NIHR or the Department of Health. The funders had no role in the study design, nor in data collection, analysis or interpretation, nor in the writing of the report or the decision to submit the article for publication. The SPHR is funded by the NIHR. NIHR SPHR is a partnership between the universities of Sheffield, Bristol, Cambridge, Exeter and UCL; the London School for Hygiene and Tropical Medicine; the LiLaC collaboration between the universities of Liverpool and Lancaster and Fuse; and the Centre for Translational Research in Public Health, a collaboration between Newcastle, Durham, Northumbria, Sunderland and Teesside universities. Natsal-3 was supported by grants from the Medical Research Council (G0701757) and the Wellcome Trust (084840), with contributions from the Economic and Social Research Council and Department of Health. CA is supported by the National Institute for Health Research (NIHR) Collaboration for Leadership in Applied Health Research and Care of the South West Peninsula (PenCLAHRC). However, the views expressed in this report are those of the authors and not necessarily those of NIHR, PenCLAHRC or the Department of Health.

**Competing interests** AMJ has been a Governor of the Wellcome Trust since 2011. All other authors have no competing interest to decline.

**Ethics approval** Work Package 1: ScHARR Research Ethics Committee granted ethical approval for this study (17.2.14). Work Package 2: Not applicable. Work Package 3: The School of Health and Social Care Research Governance and Ethics Committee at Teesside University granted ethical approval for this study (Ref: 061/14). Work Package 4: Natsal-3 was granted ethical approval by the Oxfordshire Research Ethics Committee A (Ref: 09/H0604/27). Work Package 5: Not applicable. Stakeholder consultations were with established Young Peoples Advisory Group members and professionals who had given their consent to participate.

**Provenance and peer review** Not commissioned; externally peer reviewed.

**Data sharing statement** The Natsal-3 data set (Work Package 4) is available in the UK Data Service repository, unique persistent identifier: 10.5255/UKDA-SN-7799-1;).

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
