## [Reviewer comments · BMJ Open]

ARTICLE DETAILS

TITLE (PROVISIONAL)	What is best practice in sex and relationships education? A synthesis of evidence, including stakeholders' views.
AUTHORS	Pound, Pandora; Denford, Sarah; Shucksmith, Janet; Tanton, Clare; Johnson, Anne; Owen, Jenny; Hutten, Rebecca; Mohan, Leanne; Bonell, Christopher; Abraham, Charles; Campbell, Rona

VERSION 1 - REVIEW

REVIEWER	Nancy Berglas, DrPH University of California, San Francisco USA
REVIEW RETURNED	17-Nov-2016

GENERAL COMMENTS	This is an interesting and well-written paper that considers the important question of best practices in sex and relationships education (SRE) in the UK. The authors have synthesized results from a number of different studies and data sources, which brings strength to their conclusions. Methods 1. For clarity, it would be useful to note at beginning of the Methods (top of Page 6) that the paper draws from a number of previously published analyses, that these findings are brought together with a particular purpose in mind, and that more detail about each is available in the cited literature.2. It would help to note which studies were designed to address the particular questions of how to make SRE effective, acceptable, sustainable or implemented with fidelity. (For example, the qualitative synthesis of young people's views likely addressed issues of acceptability, but not sustainability.) A simple table of the varied study methods and their purpose might be useful for readers to reference while reading through the results.3. More methodological detail is needed for the stakeholder consultations (Page 9) since these have not been published elsewhere. What is the sociodemographic profile of the participants? What types of questions were asked? How were the data analyzed? etc.4. The authors do not address the methodological limitations of their work sufficiently in the initial bullets (Page 4) and body of the paper (Page 24). It is difficult to get a sense of the limitations of each of the methods individually from the existing brief descriptions. While synthesizing s across data sources provides a number of strengths, the limitations of the overall study need to be discussed.
--

	Results 5. Even with references to the published studies, the authors need to present additional information about the individual study results to allow the reader to understand and judge their implications. Some of the summaries are too brief to be considered “results” of this paper. For example, statistical support is needed for the results presented from the Natsal data and review of systematic reviews (Pages 9-10). Additionally, there is little variation of opinion presented in the qualitative results (Page X). Were there any notable disagreements? 6. Given that a number of the studies that used qualitative methods, it is striking that the voices of young people and adult stakeholders are not included through direct quotes. Are these available? Discussion 7. I am concerned by the definitive conclusion presented in the Abstract and Discussion (Page 23) that “SRE should take place in schools.” I don’t see where this is drawn from the data. Rather, it seems that school setting is a starting point for the authors’ varied analyses (that is, that the authors set out to study school-based SRE in the UK). Other settings for SRE are not discussed and thus cannot be compared. Moreover, according to your Natsal results, 40% of young people cited school lessons as their main source of sex information, meaning that 60% cited non-school sources of information as most important. 8. Given the broad scope of the best practices presented in Table 1, the Discussion should address the feasibility of implementing these elements in UK schools. Are some elements more likely to be accepted than others? What are the anticipated barriers? What additional information is needed to move forward implementing these practices? Minor Comments 9. The paper is specific to the UK context, but is often framed as universal. This journal has many international readers. The context should be made clear in the title (e.g., “What is best practice in sex and relationships education in the UK?”) as well as in the objectives and results section of the abstract.
--	--

REVIEWER	Maria Teresa Tenconi Department of Public Health, Experimental and Forensic Medicine- University of Pavia- Pavia- Italy
REVIEW RETURNED	05-Dec-2016

GENERAL COMMENTS	The paper is clear and complete in each of its parts. I believe that it will be extremely useful to SRE discussion and planning of school programs in near future. REVIEW
--

	The paper “ What is best practice in sex and relationships education (SRE)? A synthesis of evidence, including stakeholders’view” examines the methodology and results of the most relevant researches about SRE in the last 15 years span. The Authors’ review is enriched by the consultations with stakeholders and health professionals about the feasibility of SRE, keeping in account the different points of view of young people who experienced it. In the Introduction the Authors explain clearly the background and the aims of the study. The organization in paragraphs of methodology and results facilitate their reading. In Methods the choice of studies and of their results’ analysis are wide and well documented. Concerning the Natsal-3 data, the Authors report even sample size and response rate to the survey. Results are clear and complete, keeping in account ongoing activities and underlying the importance of SRE, which should be carried on in school at different ages. Several approaches to SRE are concisely described, keeping in account either young people or professionals’ views. Furthermore the Authors describe the qualitative characteristics of a good SRE provision , their opinions about who should deliver SRE and how planning and evaluating programs. Table 1 outlines the criteria for best practice in SRE. It could be very useful to teachers and health professionals as a proper “guideline” in discussions about the organization of SRE programs for young people. Perhaps Discussion can be shortened, by omitting the summary of results at the beginning, leaving the first sentence, the strenght and limitations of the study and the Authors’ opinions about a future organization and delivery of SRE. The paper should be published in BMJ Open.
--	---

REVIEWER	Julie Bayley Coventry University, UK
REVIEW RETURNED	08-Dec-2016

GENERAL COMMENTS	Many thanks to the authors for an interesting paper which offers a useful contribution to this topic area. I very much welcome the synthesis of disparate evidence with the view to simplifying the recommendations for sex education provision. I appreciate the mixed methods applied and sense of triangulation weaved throughout the paper. Overall I am keen that this paper progresses to full publication.
--

However, I do have a series of comments and queries which need to be addressed before final acceptance. Largely speaking, these reflect a difficulty in following a narrative which pulls from multiple sources and seeks to answer multiple questions. I have identified the issues below, and in places suggested some alternative phrasing/structuring (NB I offer these as examples of how changes could be made rather than mandating a set format).

INFORMATION ON PARTICIPANTS

Throughout the paper there was very little information on each of the participant groups. For example, under WP3 (p7, lines 34-40), the authors list 'data were collected using...interviews (with LA members, school teaching staff.....informal discussions with key partners.....'. There is no information on group membership, numbers, nature of schools or who counts as 'key partners'. This issue is echoed throughout each of the sections. I recognise there is a word limit, but if necessary I would ask that this information be added with other text moved to an online appendix.

WORK PACKAGE DESCRIPTIONS

The abstract lists 5 work packages, plus stakeholder consultations. I understand that the latter method was used to 'fill in the gaps', but differentiating between work packages and 'an added bit' doesn't help the reader. It also makes it hard to follow what is new data vs. what is a review of existing published work. I would prefer these were listed as six activities with clear links to the research aims eg:

- 1) Practitioner interviews to explore best practice and barriers
- 2) Synthesis of existing qualitative studies on young people's attitudes to SRE
- 3) Qualitative case study review of existing sex education interventions to assess acceptability
- 4) Analysis of selected NATSAL data on sex education information sources and outcomes
- 5) Review of existing systematic reviews to establish effectiveness of sex education
- 6) Stakeholder consultations with young people and professionals to consolidate study findings

As these activities become data sources in the terms of the paper, it would also help to have a clear short reference explicitly stated for each which is then followed throughout the paper.

RESULTS

Following from the above, the two subheadings here – "1. Findings from work packages" and "2. Findings from stakeholder consultations" have the effect of structuring the paper chronologically rather than by the research question. It also serves to feel repetitive and harder to follow. I recommend that, especially the authors cite the paper being relevant for practitioners', that this be structured by the research questions rather than the methods, for example eg:

- 1) Importance of SRE
- 2) Effectiveness of SRE
 - a. Overall
 - b. Comparative utility of different intervention types
 - c. School based vs. school linked
- 3) SRE Content
 - a. Life skills approach
 - b. Risky behaviour approach
 - c. Abstinence approach

- d. Sex positive approach (NB the authors need to take abortion out of here or relabel the section)
- e. Cultural sensitivity approach
- 4) SRE delivery and facilitative conditions
 - a. Adaptable
 - b. Age appropriate
 - c. Spiral
 - d. Sufficient duration and intensity
 - e. Interactive and engaging
 - f. Safe
 - g. Confidential
- 5) Delivery agent choices
 - a. Teachers
 - b. Sexual health professionals
 - c. Peer educators
- 6) Planning SRE
 - a. Practicalities and pressures
 - b. Fidelity

This kind of structure – which isn't wildly different to that already there – would cut through the repetition and struggle to work out how various bits of data connect.

SECURER LANGUAGE

Throughout the paper, I found it difficult to determine the strength of evidence for the conclusions drawn. This links back to the lack of detail about sample details, meaning that phrases like 'some participants' do not reflect a depth of data. To address this, I'd ask the authors to (i) offer better detail the participant groups in the WPs, (ii) where possible, replace 'some' with more substantive terms, and (iii) consistently express data in terms of consensus/contrast across sources.

EXCLUSION OR INCLUSION OF HIV AND ALCOHOL PROGRAMMES

On page 10 lines 40-49, the authors offer commentary on the utility of HIV prevention programmes and programmes addressing alcohol and sex. However, on pg7 lines 6-7 there is a statement that HIV and alcohol programmes were excluded from the qualitative review. The rationale for this is unclear, especially when information on such programmes are then included in another work package. Can the authors clarify why there were excluded.

DISCUSSION

There needs to be a stronger statement on limitations related to participant samples. Without detail on who was involved in interviews in the various work packages, it is not possible to know what these precise limitations are, but I would expect to see here a methodological caveat relating to representativeness and generalisability based on the populations engaged

Additional specific points:

P8 line 52 – please explain or provide a reference for AMSTAR

Sex positive approach (p12 line 38 – p13 line 42). This section includes a range of important issues, but several (eg. offering abortion information and invisibility of LGBTQ students) sit awkwardly under a 'sex positive' banner in the way they are currently stated. Could this section be retitled to reflect inclusivity

	and expanding the curriculum, rather than 'positivity' per se? Ethics – please provide details of ethical approval, consenting processes and any issues relevant to disclosure and bias. "Sufficient duration" (pg20, line 54) - please define 'sufficient'
--	--

VERSION 1 – AUTHOR RESPONSE

Reviewer 1. Nancy Berglas

1. We have followed Dr Berglas' recommendations and noted at the beginning of the Methods section that we have synthesised the findings from the separate research packages to distil evidence about best practice, and that more detail about the separate research packages is available in the published reports.

2. We agreed with Dr Berglas' suggestion for a table illustrating the various aspects of programmes addressed by each study (acceptability, sustainability etc.) This information is now found in Table 1.

3. Dr Berglas asks for more methodological detail on the stakeholder consultations. However, we have treated the feedback from the stakeholder consultations differently from research data as stakeholder feedback does not constitute research data and because participants in public involvement activities such as this do not consent to have their contributions treated as research data. In our study they took part in order to give feedback on our research, not as research subjects. Consequently we did not collect detailed information on participants. However Dr Berglas' comment was helpful in making us realise that we needed to explain this and we have done so in the section entitled 'Stakeholder consultations' at the end of the Methods section.

4. Dr Berglas suggests that we need to address the limitations of the overall study in greater depth. We agree and have added some further limitations, both in the Discussion and in the bullet points (our inability to recruit parents to the case study investigation and the issue of the generalisability of our findings beyond the UK).

5. Dr Berglas asks us to present additional information about the individual study results, including statistical information where appropriate and more information on any variation of opinion in the synthesis of young people's views. We have done so in Appendix 1, which provides a summary of the findings and conclusions for each of the individual work packages, and which could be available online as supplementary information.

6. Dr Berglas asks for quotes given that some of the studies used qualitative methods. We have included a range of quotes from the three studies that used qualitative methods.

7. Dr Berglas does not agree with our conclusion that 'SRE should take place in schools'. We agree that this statement does not completely accurately reflect our findings and have deleted this from the abstract and Discussion.

8. Dr Berglas suggests that in the Discussion we consider the feasibility of implementing the elements of best practice that we outline in Table 1, giving thought to whether some elements are more likely to be accepted than others, etc. We have done as she suggests and have inserted a new paragraph to address this issue (third paragraph in the Discussion).

9. Dr Berglas writes that the paper is specific to the UK but is often framed as universal. In fact, although the studies that relied on primary data collection took place in the UK, the review of reviews and the qualitative synthesis drew on studies worldwide. We have clarified the location of studies in the Methods section where this was not already stated. However, we do not believe that the findings are only relevant to the UK so do not think it appropriate to alter the title as Dr Berglas recommends. However, we do note (in the Discussion) that our criteria for best practice will probably be of the greatest relevance in a UK setting.

Reviewer 2. Maria Teresa Tenconi

This reviewer had no comments except to say that the paper was clear and complete and that it would be extremely useful for the planning of school programmes.

Reviewer 3. Julie Bayley

Dr Bayley is very positive about the paper and makes a number of suggestions:

1. She recommends including more information about study participants in each of the work packages. We agree and have added further information about participants in the primary studies where necessary. We have also added more information about studies included in the evidence syntheses.
2. Dr Bayley suggests that we should not differentiate between the five work packages and the stakeholder consultations; she suggests that treating them as six activities would make it easier to follow the narrative. However, as noted above (see point 3, Reviewer 1) stakeholder contributions do not constitute research data and for that reason we feel it is important to maintain the distinction between the research and the stakeholder consultations.
3. Dr Bayley recommends that the work packages are linked to their aims. We agree and have already addressed this point by creating Table 1 for this purpose, as recommended by Reviewer 1 (point 2).
4. Dr Bayley suggests referring to each of the activities (the work packages and stakeholder activities) using a clear short reference throughout the paper. We did try this in an earlier version of the paper, referring to WP1, WP2 etc. (work package 1, 2 etc.) but found that it didn't flow as well as the current version since the reader had to keep referring back to a table to remind herself which study was being referred to.
5. Dr Bayley suggests that the paper is organised differently to make it easier to follow the narrative. She recommends, as above, that we should not differentiate between the work packages and the stakeholder consultations. However, for reasons already discussed, we believe that it is important to keep these two activities distinct, so we have not reorganised the paper as she suggests.
6. Dr Bayley suggests that we clarify the strength of evidence for the conclusions drawn by giving more detail about the participants, by using more substantive terms and consistently expressing data in terms of consensus/ contrast across sources. As noted above, we have included more detail about the participants in the Methods section. We have also attempted to clarify the strength of evidence where necessary. However, we feel that we have been clear in terms of expressing consensus and disagreement where this exists.
7. Dr Bayley asks why the qualitative synthesis excluded studies on HIV and alcohol, while the review of reviews included them. In fact we stated that the qualitative synthesis excluded studies that only focused on HIV and alcohol (i.e. in Methods section, no. 2: 'Studies were excluded if they focused solely on alcohol or HIV/ AIDS ...'). Consequently, if a qualitative study was about sex and relationship education and included aspects about HIV and alcohol, it was eligible for inclusion.
8. Dr Bayley asks for a stronger statement on the limitations of the study in relation to the samples in each study, paying particular attention to issues of representativeness and generalisability. However, we do not believe that any of the individual study samples suffer from limitations that compromise their generalisability (where generalisability is an appropriate concern). The telephone interview study gained representation from every English local authority, as it aimed to do; studies included in the qualitative synthesis were obtained using comprehensive electronic and hand searching strategies; the case study investigation identified appropriately diverse interventions using purposive sampling; NATSAL selects households using stratified probability sampling; the review of reviews used comprehensive electronic search strategies. We believe this information is clearly explained in the Methods section but we have added additional clarification where necessary. We have however, noted in the Discussion that although the evidence syntheses drew on data from all over the world, our primary research data is confined to the UK. Consequently, we note that while the international data supports and does not contradict our UK based findings, our criteria for best practice probably have greatest relevance in a UK setting.
9. Dr Bayley asks that we explain or provide a reference for AMSTAR. We have inserted the

appropriate reference in the text (Shea et al. 2007).

10. Dr Bayley questions whether 'sex-positive' is the appropriate heading for a section that includes abortion information and LGBTQ issues. We argue that it is appropriate since sex-positive SRE would reflect and embrace sexual diversity and would be comprehensive, i.e. would include all the information necessary for young people to make choices (such as all the options available in the event of pregnancy).

11. Dr Bayley asks us to provide details of ethical approval, consenting processes and any issues relevant to disclosure and bias. Details of ethical approval for each of the studies has already been submitted in the online submission process with ScholarOne Manuscripts. Ethics committees will have considered the appropriateness of consenting processes when they granted approval for the studies. Issues relevant to disclosure and bias are already given in the Declarations statement.

12. Dr Bayley asks us to define 'sufficient' as it relates to 'sufficient duration' in what is now Table 2. We have clarified this.

Additional review (anonymous, sent as an attachment)

This reviewer feels that the paper is well organised, easy to read and that the method and results are clear, complete and concisely described. The only suggestion is to shorten the Discussion by omitting the summary of results at the beginning. However, BMJ Open ask for a summary of findings as the first paragraph of the Discussion so we have retained this.

VERSION 2 – REVIEW

REVIEWER	Nancy F. Berglas, DrPH University of California, San Francisco USA
REVIEW RETURNED	06-Feb-2017

GENERAL COMMENTS	Thank you for the opportunity for review this paper again. The question of best practices in sex and relationship education is a critical one, and the authors do a commendable job synthesizing information from multiple data sources. My primary concerns – regarding the detail of each method, addressing feasibility, and acknowledging the limitations of the study– have been addressed. I recommend this paper for publication, with the following minor suggestions. Abstract: 1. There is still need to specify the setting (U.K., England, etc). This is done in the paper, but needs to be clear in the abstract. For example, the objectives should read “but in England SRE is not statutory, a third of schools lacks...” because this context differs across countries. 2. The results need to be presented as data drawn from this particular study. As is, they are over-interpreted. I suggest this section begin with “Using multiple data sources, we find that...” or “Results from this study indicate that...” Additionally, I recommend use of the past tense (“agreed” as opposed to “agree”) to make clear that these are the views that were reported to you by study participants, rather than broad conclusions: “We found professional consensus that...”, “Young people and professionals agreed that...”, “When surveyed, young people advocate a ‘sex-positive’ approach...” Methods:
---

	3. I understand that the stakeholder information was not collected as research. I have participated in similar activities. However, by including it here in the paper, it is being presented as research – irrespective of the availability of participant details. It is still not clear how the research findings from the work packages were shared with stakeholders (presentation? reports?), how participant response was collected (recordings? note taking? surveys?), what questions were asked, how the data were summarized, etc. More detail is still needed. Appendix: 4. This is useful information, but difficult to distill and missing methods. I suggest adding a column on the methods, its limitations, a briefer/bulleted version of the results, and citations to published version of these results. 5. The link to the appendix should be placed near the results, so readers are aware as they go through that section where they can find more. Other comments: 6. The acronym PSHE is never written out. As a non-UK reader, I do not know what this is. LA is defined at first used, but I kept having to look back when it reappeared. Could “Local Authorities” be written out to assist with international readers? 7. I agree with the third reviewer’s comment that the narrative of the results is challenge to follow because of multiple data sources. I strongly support the reviewer’s recommendation to include main headers (Importance of SRE, Effectiveness of SRE, SRE Content, etc.), in bold or numbered. This does not change the text, but would be very useful in guiding the reader through the volume of text. 8. Approval or exemption by an institutional review board needs to be included.
--	--

REVIEWER	Julie Bayley Coventry University, UK
REVIEW RETURNED	02-Feb-2017

GENERAL COMMENTS	I am happy to recommend this paper be accepted for publication. The authors have made a series of very useful edits (as per comments by all reviewers) and I feel it is now both clearer and more comprehensive in its content. I appreciate the authors' willingness to integrate the comments and am happy with the decisions they have made in terms of my suggestions (both where accepted or declined). I remain - personally - uncomfortable with the use of the term 'sex positive', but am also aware that (i) this is because I associate positive in this area with 'pleasure', and thus reflects my preferred language rather than a hard line definition, and (ii) the definition the authors give is comprehensive and justifiable. As such, the authors' decision to retain it is justified.
--

	An interesting paper, which I hope will support discussion and decision making around sex education design/implementation.
--	--

VERSION 2 – AUTHOR RESPONSE

Response to reviewer Nancy Berglas's comments

1. Dr Berglas asks us to specify the setting in the abstract. We have done so.
2. Dr Berglas suggests we rewrite the results section of the abstract to put them in the past tense and to avoid over-interpretation. We have done so.
3. Dr Berglas asks us to include more information on the stakeholder consultations. As she suggested we have included more information on how the research findings were shared with consultees, how consultees' responses were recorded, what questions were asked and how the information was summarised. We have also referred to our unpublished reports on these consultations.
4. Dr Berglas asks us to add a column to the table in the Appendix for methods and to edit the information in the results column. She also asks us to add citations to relevant published papers. We have added a column on methods but have provided brief information only as full details of methods are given in the text of the paper itself. We have edited the results column so that this is now much shorter and have added a column for published papers.
5. Dr Berglas asks us to refer to the Appendix near the Results section so that readers know where they can obtain more information as they go through the results. We have done as she asked and inserted a reference to the Appendix at the end of the Methods section, just before the Results section, but are unsure whether this works as we also refer to the Appendix at the beginning of the Methods section.
6. We have written out Personal, Social and Health Education where it first appears, to explain the acronym PSHE, as Dr Berglas suggests. We have also written out 'local authorities' wherever used, instead of using the acronym, as Dr Berglas asks.
7. Dr Berglas suggests that we put the main headers in bold or numbered. As we formatted the headings according to BMJ Open's manuscript requirements we were unsure whether to change them, so have left them as they are.
8. Dr Berglas asks us to provide details on institutional review board approval/ exemption. Full information on ethical review for each of the studies has been submitted but this is online and not visible on the manuscript itself at this stage.

We really appreciate the time and attention that Dr Berglas has given to this paper. We have made some additional changes as follows:

- We have edited the abstract, introduction, p21 (second para) and conclusion in view of the government's recent announcement of their intention to make SRE statutory in all English schools.
- When discussing the benefits of external professionals as sex educators we have referred to their 'separateness from the school' rather than their anonymity, as on reflection this is more accurate. (p20, second para, and in Table 2.)
- We have made various other small edits where necessary to increase accuracy and clarify meaning (e.g. Discussion: first para).

VERSION 3 – REVIEW

REVIEWER	Nancy F. Berglas, DrPH University of California, San Francisco
REVIEW RETURNED	15-Mar-2017

GENERAL COMMENTS	My concerns have been addressed through the review process. I recommend this paper for publication.
---